# Co-Inoculation of *Bradyrhizobium* spp. and *Bacillus* sp. on Tarwi (*Lupinus mutabilis* Sweet) in the High Andean Region of Peru

**Mariela Monroy-Guerrero [1], Miriam Memenza-Zegarra [1], Nataly Taco [1], Elvia Mostacero [2], Katty Ogata-Gutiérrez [1], Amelia Huaringa-Joaquín [2], Félix Camarena [2] and Doris Zúñiga-Dávila [1,\*]**

[1] Laboratorio de Ecología Microbiana y Biotecnología, Departamento de Biología, Facultad de Ciencias, Universidad Nacional Agraria La Molina (UNALM), Lima 15024, Peru

[2] Programa de leguminosas, Facultad de Agronomía, Universidad Nacional Agraria La Molina (UNALM), Lima 15024, Peru

\* Correspondence: dzuniga@lamolina.edu.pe

**Abstract:** Tarwi (*Lupinus mutabilis* Sweet) is an Andean legume that has attracted international interest due to its high nutritional value. This has resulted in an increase in its conventional production, which leads to an ecological imbalance. In this context, the application of biotechnologies, based on the use of bacterial inoculants, is of utmost importance. This work aimed to evaluate the effects of a consortium of 2 strains of *Bradyrhizobium* spp. (BR) and 1 strain of *Bacillus* sp. (BA) on tarwi var. Andenes. The treatments tested were BR + BA, BR + Organic Matter, BR + Agrochemical (Azoxystrobin y Difenoconazole), and the control (without application). The crop was located in Marcara-Ancash (altitude 3254 masl), Peru. The experiment involved the inoculation of BR in the seeds and a re-inoculation 30 days later. BA was inoculated every 30 days in the neck of the plant and aerial part, 5 times during plant development. The inoculation with BR + BA significantly increased the aerial fresh weight (413.2%), plant height (13.5%), and diminished the anthracnose (38.4%) of plants 110 DAS (days after the sowing). Also, this treatment produced the best-harvested emergence percentage (97.9%), morpho-agronomic characteristics, and an increase in the yield (171%) compared to the control. In conclusion, the application of the *Bacillus* sp. strain and the *Bradyrhizobium* spp. consortia improved the productivity of tarwi var. Andenes. The interaction of these strains have the potential to be used in tarwi field planting programs.

**Keywords:** *Bradyrhizobium* spp.; *Bacillus* sp.; biotic interaction; plant-growth promoting bacteria

## 1. Introduction

Tarwi is a legume that is grown between 2000 and 3800 masl (meters above sea level) [1]. Its seeds are appreciated for having a high protein and oil content [2]. For these reasons, in the last few years in Peru, tarwi production has increased, reaching yields of 1250 kg/ha in 2020, compared to the 850 kg/ha registered in 2019 [3]. On the other hand, one of the main fungal diseases of tarwi is anthracnose caused by *Collettrichum* spp. [4]. Usually, this disease is treated with chemical products, which entail a high cost of production and an ecological imbalance. Moreover, the use of plant breeding has not improved its production, reducing even its resistance features in some cases. A sustainable alternative for these problems is the use of symbiotic bacteria such as *Bradyrhizobium* sp., which actively participates in biological nitrogen fixation (BNF) on legumes [5], and a biocontrol bacterium such as *Bacillus* sp., that is reported for its biocidal activity against different species of phytopathogenic fungi [6]. For example, *Bacillus* sp. B02 reduces the growth of phytopathogenic fungi through the production of antimicrobial peptides, siderophores, and hydrolytic enzymes such as cellulases and proteases causing the lysis of cell wall components [7]. At field conditions, the same bacterium reportedly reduced the

incidence of the disease caused by *Sclerotinia sclerotiorum* by 94%. Therefore, the work aimed to evaluate the effects of the inoculation of a consortium of two strains of *Bradyrhizobium* spp. LMHZ L8 and LMHZ L3 (BR) and 1 strain of *Bacillus* sp. B02 (BA) on tarwi growth and production. For that reason, morpho-agronomic characteristics such as aerial fresh weight, nodules, chlorophyll content, anthracnose severity degree (during flowering); grain yield, nutrient content, and percentage emergence of seeds obtained at the end of the harvest were evaluated.

## 2. Materials and Methods

### 2.1. Experimental Field Location

The experimental field was located on Ancash department, Carhuaz province, Marcara district in Tuyu Alto Zone at 3254 masl (Supplementary Figure S1) (Latitude 9°18′39.17″ S; Longitude. 77°34′29.39″ W) (Supplementary Figure S2 and S3). The research was carried out during the 2019–2020 campaign. During the experiment, precipitation varied from 42.19 mm in February and March to 0 mm in August. The maximum temperature recorded was in the range of 18.3 to 19.37 °C and the minimum temperature was between −3.95 to 2.28 °C, with relative humidity ranging from 55.8% to 71.56% [8] (Supplementary Figure S4). According to Senamhi, the climate can vary from semi-arid when the weather is temperate to semi-dry in the winter [9].

### 2.2. Tarwi Seeds Var. Andenes

The tarwi var. Andenes comes from the Junin area. This variety is precocious and has a cycle of 5 to 6 months, however, its phenological cycle depends on the climate. The stem is not prominent and herbaceous, with a maximum height of 1.2 m. The flowers are light pink, and the grains are white and round. It has 17 pods per central inflorescence and a maximum of 53 pods in the lateral inflorescences. Under optimal conditions, it achieves an average yield of 2950 kg/ha. It was also reported to be susceptible to rust and anthracnose [10]. It was reported that the Andenes variety possessed a protein content of 42.2 g/100 g of seeds, highlighting its oil content with 18.6 g/100 g of seeds [11].

### 2.3. Bacterial Strains and Growth Conditions

The bacteria used in this experiment were obtained from the Bacteria Culture Collection of the Laboratorio de Ecología Microbiana y Biotecnologia of the Universidad Nacional Agraria, La Molina (Lima, Peru).

#### 2.3.1. Bradyrhizobium spp. Consortium (BR) Inoculant

Strains LMHZ L8 and LMHZ L3 were isolated from tarwi nodules, obtained from the cultivated fields of Huaraz, in the Ancash department of Peru. The soil feature was loamy clayey and partially acidic (pH 4.5–6.5). The strains were characterized as *Bradyrhizobium* spp. in previous work using microbiological and molecular methods [12]. *Bradyrhizobium* spp. strains were able to perform biological nitrogen fixation in symbiosis with the legume, but they did not demonstrate other PGPR (Plant Growth Promoting Rhizobacteria) properties, nor antagonistic activity against phytopathogenic fungi such as *Fusarium* spp. *Rhizoctonia* spp. and *Colletotrichum* spp.

The consortia was made up of two strains of *Bradyrizobium* spp. LMHZ L8 and LMHZ L3 [5], which were grown separately, in 250 mL of YMB (Yeast Mannitol Broth) medium. The culture was incubated at 28 °C for 5 days on a rotating incubator (MRC, Israel) with continuous stirring at 150 rpm. For field inoculation, bacteria are required to achieve a concentration of >1 × 10$^8$ CFU/mL. For the consortium treatment preparation, 5 mL of each bacterial culture was mixed. Then, 10 mL of the inoculant was incorporated into 200 g of agricultural soil (field soils) and the whole mixture was homogenized with 1 kg of tarwi seeds. Pelleted seeds were left to dry in shade until sowing. To guarantee bacteria colonization, a re-inoculation was performed using a dilution of 10 mL of the consortium

on 1 L of non-chlorinated water. The inoculant was applied to the region of the plant's neck, 30 days after sowing (DAS).

### 2.3.2. *Bacillus* sp.(BA) Inoculant

*Bacillus* sp. was isolated from common bean plants grown in the coastal desert of Ica, Peru [13].To isolate *Bacillus* sp., the soil sample was pre-treated in a water bath for 30 min at 80 °C. The samples were then serially diluted up to $10^5$ and 1 mL of sample was incorporated into a TGEA (Tryptone Glucose Yeast Extract Agar) medium. The incubation was carried out at 28 °C for 24 to 48 h [13]. Molecular identification and phylogenetic analysis were performed by amplifying 16S rRNA genes and Sanger sequenced at Macrogen. Molecular identification analysis was performed by comparing DNA sequences against the sequences of strains retrieved for GenBank. Phylogenetic analysis was performed by neighbor-joining (NJ) and genetic distances were computed with Kimura's two-parameter model using Mega7 [13].

The *Bacillus* sp. strain BO2 [7], was grown in 250 mL of MMM (Minimum Mineral Medium). The culture was incubated at 28 °C for 3 days on a rotating incubator (MCR, Israel) with continuous stirring at 150 rpm. For field inoculation, the strain was grown to a concentration of $>1 \times 10^8$ CFU/mL. The inoculant was then transported in a cooler with gel packs to conserve its integrity. For field application, the inoculant was mixed in water without chlorine in a 20 L backpack sprayer. The proportion of the components used in the mixture was prepared according to what is indicated in Table 1. The inoculant was applied to the entire plant (leaves and neck regions of the plant), following the established dose (Table 1) for each DAS (days after sowing).

**Table 1.** Dosage of *Bacillus* sp. inoculum in tarwi plants.

| Number of Doses | DAS | mL of Inoculum/1 L Water * |
|:---:|:---:|:---:|
| 1 | 60 | 15 |
| 2 | 90 | 30 |
| 3 | 115 | 90 |
| 4 | 150 | 90 |
| 5 | 180 | 120 |

* Non-chlorinated water.

### 2.4. Treatments and Their Distribution

The experiment was set up in a field structured into four treatment groups (Table 2), each having three blocks (repetitions). A Completely Randomized Block Design (DBCA) was used, with experimental units organized in plots and blocks (Figure 1). The plants were planted at 1 m$^2$ spacing. The experiment comprised 3 rows. A total of 36 plants were sown per row, considering that 4 seeds were sown per hole at a spacing of 0.30 m.

During the experiment, all cultural work was carried out on tarwi crop var. Andenes. Irrigation depended entirely on rainfall because the experiment was located in a high area without access to a water source. Soil samples for physicochemical analysis were sent to the testing laboratory for soils, plants, water, and fertilizers at Universidad Nacional Agraria La Molina. The analysis samples were air-dried and sieved in a 2 mm mesh sieve. Sand, silt, and clay percentages were determined using the hydrometer method. Soil salinity was determined by measuring the electrical conductivity of the liquid extract in a 1:1 soil/water ratio. The pH was established in a ratio 1:1 (soil:water) using a potentiometer. The common gas–volumetric method was used to determine carbonate ($CaCO_3$). The organic matter (OM) was measured by the Walkley and Black method (potassium dichromate oxidation). The available phosphorous was extracted from the soil with 0.5 M $NaHCO_3$ solution adjusted to pH 8.5 using a procedure modified from Olsen method. The K+ exchangeable base was determined by the extraction with $NH_4OAC$ adjusted to pH 7.0 +followed by quantification using atomic absorption [13].

**Table 2.** Treatments used in the experimental trial.

| Treatments | Features | Inoculant Application |
|---|---|---|
| *Bradyrhizobium* spp. + *Bacillus* sp. (BR + BA) | *Bacillus* sp. [5] strain B02 (BA) and a bacterial consortium with strains: *Bradyrhizobium* sp. [4] (LMHZL8 + LMHZ L3) (BR). | *Bradyrhizobium* spp. was applied when sowing and 30 DAS. *Bacillus* sp. was applied according to Table 1 |
| *Bradyrhizobium* spp. + Organic Matter (BR + OM) | Organic Matter (OM) composted cow manure (pH: 8.62; Nt%: 1.421; P%: 0.266; K%: 1.031; C.E. dS/m: 2.30) | *Bradyrhizobium* spp. was applied when sowing and 30 DAS. Organic matter was applied manually at sowing (Dose: 2 kg/row) |
| T3 = *Bradyrhizobium* spp. + Chemical control (BR + CQ) | Agrochemicals (CQ): *ConfieePlus 500 SC* (Neoagrum, China) (Azoxystrobin y Difenoconazole) | *Bradyrhizobium* spp. was applied when sowing and 30 DAS. Agrochemical was applied at the beginning of flowering (about 90 DAS), (Dose: 0.4 L/ha)/every 15 days, twice |
| T4 = Control | No applications | - |

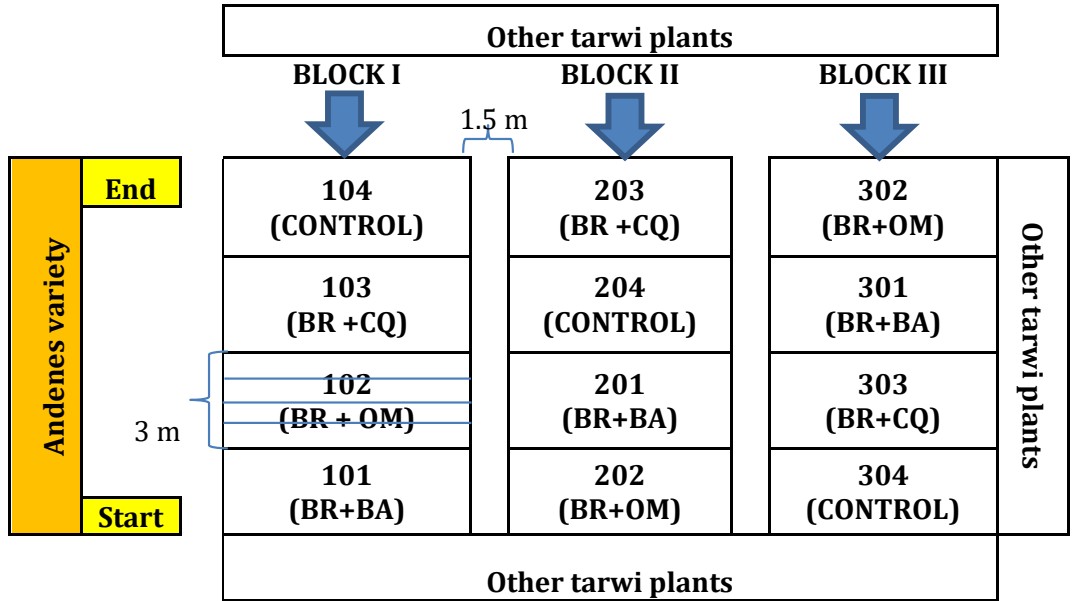

**Figure 1.** Distribution of treatments in the field. The 3 blue horizontal lines are the 3 rows.

*2.5. Evaluated Variables*

2.5.1. Chlorophyll Content

Chlorophyll content was indirectly evaluated using the SPAD-502 Plus (Konica Minolta, Alemania), reading checker 69.0 ± 3.0, which is expressed in SPAD degree [14]. The evaluation was made during the crop flowering stage. The SPAD-502 quantitatively evaluates the green intensity of the leaf, measuring light transmissions at 650 nm, where light absorption by the chlorophyll molecule occurs, and at 940 nm where no absorption occurs.

2.5.2. Aerial Plant Weight and Vigor (g)

These parameters were evaluated during the flowering stage, for this purpose, three plants per treatment per block were sampled. To determine the aerial plant weight, leaves and stems of plants were sampled and weighed using a portable digital scale (Camry, China), maximum weight 5 kg, with decimal 0.1 g). To determine the vigor of the plants a scale table (Table 3) was used and compared with observed plants features.

**Table 3.** Vigor scale.

| Amount | State | Plant Appearance |
|--------|-------|------------------|
| 1 | Very bad | Very rickety plant, very chlorotic, with very few leaves, some even dead. |
| 2 | Bad | Rickety plant, chlrotic and with few leaves. |
| 3 | Regular | Plants without rickety or chlrotic with few leaves. |
| 4 | Good | Plants with presence of development, with regular leaves. |
| 5 | Very good | Plants with the presence of greater development and abundant presence of leaves. |

Source: Gonzales and Chow, 2008 [15] (modified).

### 2.5.3. Characteristics of the Nodules

The nodules of three plants per plot, per block of the tarwi crop var. Andenes were collected. Nodules of each plant were counted and features such as color, shape, and size were recorded, according to the CIAT manual [16].

### 2.5.4. Anthracnose Severity Percentage

Anthracnose disease caused by the phytopathogenic fungus *Colletotrichum* spp. was evaluated at 60, 110, and 243 DAS. The damage caused by the fungus on tarwi plants was measured using the scale table (Table 4) [17]. To confirm if the pathogen is *Colletotrichum* spp., tarwi leaves with disease symptoms were sampled and placed in 2 oz. Whirl Pack bags. Bags were transferred in a cooler with gel packs to the laboratory for the fungus morphological identification. In the laboratory, leave samples were incubated in a moisture chamber following the methodology described by Falconi [17]. Then, leaves were cultivated on PDA (Potato Dextrose Agar) plates and incubated for 5 days at 26 °C [17]. After this time spores were present on symptomatic tissues and collected in microtubes with sterilized distilled water. Five serial dilutions of the collected spores were made and cultured on PDA medium, stored at 26 °C for 5 days [17]. The diagnosis was completed with the fungus and spore examination under the stereoscope and the microscope. Representative colonies of *Colletotrichum* spp. as well as the presence of ovoid conidia and spores that are tapered on one side and round on the other were observed.

**Table 4.** The scale of the severity degree and percentage of the disease caused by Anthracnose in the tarwi crop.

| Severity Degree | % | Damage Appearance |
|-----------------|---|-------------------|
| 0 | 0 | Plants without any injury |
| 1 | 20 | Very small lesion (5 mm) on leaves and stems, some wrinkling of leaves, no sporulation. isolated lesions |
| 2 | 40 | Local infections in leaves and stems. 1 cm to 0.5 cm lesions, little sporulation |
| 3 | 60 | Frequent infections in leaves, stems, and/or pods, accompanied by necrotic tissue (sporulation) |
| 4 | 80 | Presence of large lesions (more than 3 cm) on stems, branches, and pods with necrotic tissue, accompanied by tissue collapse (sporulation). Bent terminal buds |
| 5 | 100 | Severely affected necrotic plant, dead plants, form small pods, sporulation of salmon-colored tissue |

Source: Falconi, 2012 [4].

### 2.5.5. Variables Evaluated

At harvest, variables such as plant height and the number of branches of all plants were quantified. To obtain other morpho-agronomic characteristics like pod number, weight (g), length (cm), width (cm) and thickness (cm), number of seeds, length (cm), thickness (cm), and the locules number of central and lateral inflorescences, a total of 20 plants per

treatment per repetition were chosen randomly. Moreover, 15 days after the harvest, dry weight of all plants per treatment per repetition were recorded.

### 2.5.6. Yield

Evaluation of the tarwi yield was recorded after harvest. Yield was measured by using the dry weight of all the grains obtained from each treatment of the tarwi crop var. Andenes, 243 DAS. Yield values were expressed in kg/ha.

### 2.5.7. Nutrient Content of the Grain

The content of macro and micro-nutrients were analyzed after harvest. For these analyses, a composite sample of each treatment's repetition was made. Then, for each treatment, 100 g of seeds were weighed twice and sent to the Laboratorio de suelos of the Universidad Nacional Agraria La Molina, Lima-Peru.

### 2.5.8. Evaluation of the Emergence Percentage of Grains

The viability of harvested grains was evaluated by measuring the percentage of seed emergence. For this purpose, a total of 100 seeds per treatment were allowed to germinate in pots using a substrate made of sand, humus, and agricultural soil in 1:1:1 proportion. The substrate has a medium percentage of organic matter (2.5%), an acidic pH (5.5), very slightly saline (0.19 dS/m), and an absence of carbonates with a medium content of phosphorus (7 ppm) and potassium (100 ppm). In this substrate, germination testing was accomplished by using 4 pots with 25 seeds, and 4 replications per treatment; considering a completely randomized design. The experiment was located in Concepcion, Province of Concepcion, District of Concepcion. Seedlings were grown in a naturally illuminated nursery at a minimum temperature of $-1.41$ °C and a maximum temperature of 20.33 °C with 80.81% of relative humidity [8], Irrigation was carried out with an atomizer with non-chlorinated water, watering between 200 and 600 mL per day, for 3 times. After 15 days, data regarding the fresh and dry weight of the seedlings were obtained and the percentage of seedlings emergence was calculated. A total of 262 seedlings were analyzed in this assay.

### 2.6. Statistical Analysis

It was carried out through an analysis of variance with the Statgraphics Centurion 16.1 version program. The differences between the mean values of the treatments were determined by the LSD test with $p \leq 0.05$. The type of multiple comparisons the *p*-value uses the Bonferroni method.

## 3. Results

### 3.1. Physicochemical Characterization of Agricultural Soil

The physicochemical characterization of the field showed clay loam texture, medium percentage of organic matter (2.03%), strongly acidic pH (4.68), very slightly saline (0.19 dS/m), absence of carbonates, with a medium content of phosphorus and potassium, and a CEC (Cation-Exchange capacity) of 12.48.

### 3.2. Aerial Fresh Weight, Chlorophyll Content, and Vigor

During the flowering stage, plants inoculated with the *Bradyrhizobium* spp. consortium (BR) showed the capacity to improve plant aerial fresh weight, vigor, and chlorophyll content (Table 5) compared to the control.

### 3.3. Nodules

Plants inoculated with the *Bradyrhizobium* spp. consortia showed nodules with the best features. Treatment BR + BA had a higher root fresh weigth with nodules (41.72 g), nodule fresh weight (0.258 g), number (3.33) and size (4.26 mm) and internal red color compared to the control; which showed the lowest root fresh weight with nodules (15.55 g)

and only nodules fresh weight (0.017 g), the smallest size (1.57 mm) and internal whitish color (Table 6).

**Table 5.** Effect of treatments on foliar characteristics during flowering.

| Treatments | Aerial Leaf Fresh Weight (g) | Vigor | Chlorophyll Content (S.I.) |
|---|---|---|---|
| BR + OM | 375.33 (±36.55) a | Very good | 51.17 (±2.12) a |
| BR + BA | 297.67 (±64.63) ab | Good | 49.56 (±2.35) a |
| BR + CQ | 236.0 (±44) b | Good | 51.02 (±2.31) a |
| CONTROL | 58 (±14) c | Regular | 47.00 (±2.91) b |

Different letters mean statistically significant differences (value $p \leq 0.05$) in LSD's Test; (± Standard deviation). S.I.: SPAD Index.

**Table 6.** Effect of treatments on nodular characteristics during flowering.

| Treatments | Root Fresh Weight with Nodules (g) | Nodules Fresh Weight (g) | NoduleSize (mm) | Number of Nodules | Internal Color |
|---|---|---|---|---|---|
| BR + OM | 43.70 (±4.76) a | 0.606 (±0.729) a | 7.72 (±7.50) a | 4.33 (±3.21) a | Red |
| BR + BA | 41. 72 (±4.63) a | 0.258 (±0.285) a | 4.26 (±1.09) a | 3.33 (±0.58) ab | Red |
| BR + CQ | 40.38 (±2.62) a | 0.378 (±0.258) a | 8.82 (±2.58) a | 1.33(±7.50) ab | Red |
| CONTROL | 15.55 (±5.54) b | 0.017 (±0.029) a | 1.57 (±2.72) a | 0.67 (±7.50) b | Whitish |

Different letters mean statistically significant differences (value $p \leq 0.05$) in LSD Test; (± Standard deviation).

*3.4. Anthracnose Severity Degree Caused by Colletotrichum sp.*

Results showed that BR + CQ treatment significantly reduced the degree of anthracnose severity by 9.12 % with respect to the control, 60 DAS. However, when plants were evaluated 110 and 243 DAS, BR + BA (10.3% and 66.9%) and BR + OM (12.5% and 68.3%) treatments significantly reduced the degree of anthracnose severity compared to the control. Moreover, 243 DAS, treatments BR + BA and BR + OM showed better results in 5.2% and 3.7% respectively compared to the BR + CQ treatment and in 11.2% and 9.7% respectively compared to the control (Table 7).

**Table 7.** Effect of treatments on the percentage of Anthracnose disease.

| Treatments | % Anthracnose (Days after Sowing) | | |
|---|---|---|---|
| | 60 | 110 | 243 |
| BR + OM | 17.58 (±3.86) ab | 12.54 (±0.44) a | 68.39 (±7.04) a |
| BR + BA | 13.89 (±13.88) ab | 10.36 (±5.43) a | 66.95 (±5.64) a |
| BR + CQ | 10.29 (±10.28) a | 12.30 (±1.78) a | 72.10 (±0.69) ab |
| CONTROL | 19.41 (±19.41) b | 16.81 (±6.50) a | 78.13 (±2.27) b |

Different letters mean statistically significant differences (value $p \leq 0.05$) in LSD's Test; (± Standard deviation).

These results are reflected in Figure 2, where the differences between a plant inoculated with BR + BA and the control are observed, which presented less branching, less leaf mass, less height, leaf chlorosis and less inflorescence, and in turn was affected by the anthracnose.

*3.5. Grain Yield*

Results showed that the grain yield of tarwi var. Andenes significantly increased with treatments BR + BA (465.5 kg/ha) and BR + OM (466 kg/ha) compared to the control (171.8 kg/ha) (Figure 3).

*3.6. Effects on Morpho-Agronomic Characteristics*

The plants treated with the BR + BA treatment presented greater height, weight of the foliar area (stems and leaves), and branching, compared to the control, this was evidenced after 243 DAS (Table 8).

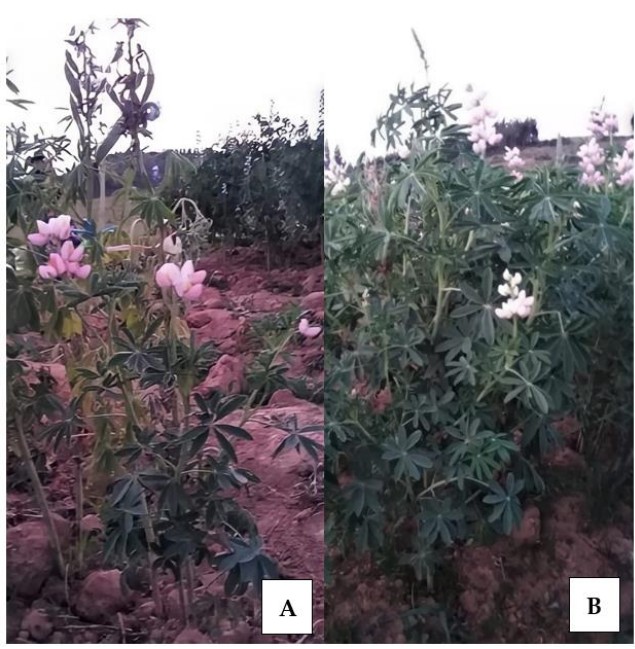

**Figure 2.** Tarwi plants var. Andenes evaluated 110 DAS. (**A**) Control; (**B**) BR + BA treatment.

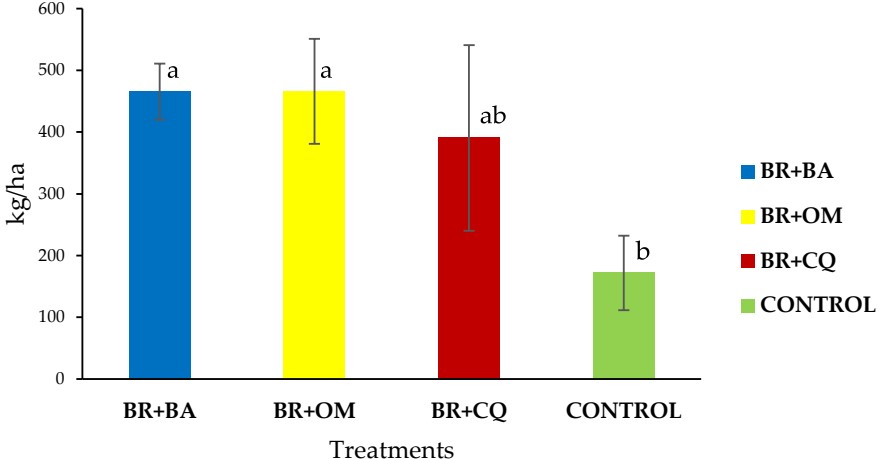

**Figure 3.** Effects of the treatments on grain yield. Different letters mean statistically significant differences (value $p \leq 0.05$) in LSD test.

**Table 8.** Effect of treatments on height, Aerial dry weight, and number of branches.

| Treatments | Height of Plant (cm) | Aerial Dry Weight (Stems and Leaves) (g) | Number of Branches (Per Plant) |
|---|---|---|---|
| BR + OM | 88.30 (±2.64) ab | 33.53 (±7.23) ab | 6.60 (±0.12) a |
| BR + BA | 84.58 (±3.72) a | 37.69 (±1.94) a | 6.73 (±0.84) a |
| BR + CQ | 81.43 (±1.38) b | 27.53 (±8.93) c | 6.32 (±0.08) a |
| CONTROL | 74.55 (±2.91) c | 29.33 (±7.64) b | 5.74 (±0.64) a |

Different letters mean statistically significant differences (value $p \leq 0.05$) in LSD Test; (± Standard deviation). Statistics analysis from plants evaluated 243 DAS.

The treatments inoculated with *Bradyrhizobium* spp. showed the best quantitative characteristics on the central inflorescence. Data highlights of the BR + BA treatment are number of pods, weight, length, and thickness at 19%, 49.8%, 9.1%, and 19.3% respectively compared to the control. Additionally, BR + BA treatment also showed the best quantitative characteristics on the lateral inflorescence too. Number of pods, weight of pods, and number of locules increase to 165.1%, 116.9%, and 14% compared to the control (Table 9)

**Table 9.** Effect of treatments on the pods and seeds of the central and lateral inflorescence.

| | CENTRAL INFLORESCENCE | | | | | | | | |
|---|---|---|---|---|---|---|---|---|---|
| **Treatments** | **Pods/Plant** | | | | | **Seed/Pods** | | | **Number of Locules** |
| | **Number** | **Weight (g)** | **Length (cm)** | **Width (cm)** | **Thickness (cm)** | **Number** | **Length (cm)** | **Thickness (cm)** | |
| BR + BA | 13.09 (±1.06) a | 22.04 (±1.54) a | 8.605 (±0.51) a | 1.453 (±0.07) a | 0.576 (±0.01) a | 5.016 (±0.46) a | 0.853 (±0.08) a | 0.323 (±0.03) a | 5.583 (±0.45) a |
| BR + OM | 14 (±0.18) a | 23.09 (±0.20) a | 9.213 (±0.46) ab | 1.496 (±0.02) a | 0.563 (±0.01) a | 5.25 (±0.26) a | 0.85 (±0.05) a | 0.326 (±0.02) a | 5.667 (±0.29) a |
| BR + CQ | 13.27 (±1.16) a | 21.62 (±4.20) a | 8.513 (±0.78) ab | 1.426 (±0.08) a | 0.526 (±0.01) ab | 4.65 (±0.57) a | 0.843 (±0.02) a | 0.31 (±0.01) a | 5.400 (±0.53) a |
| CONTROL | 11 (±1.33) b | 14.71 (±3.02) b | 7.890 (±0.42) b | 1.420 (±0.03) a | 0.483 (±0.05) b | 4.15 (±0.33) a | 0.793 (±0.04) a | 0.306 (±0.01) a | 5.033 (±0.60) a |
| | LATERAL INFLORESCENCE | | | | | | | | |
| BR + BA | 23.62 (±8.06) a | 24.13 (±4.16) a | 7.846 (±0.37) a | 1.417 (±0.01) a | 0.470 (±0.02) a | 4.173 (±0.28) a | 0.767 (±0.01) a | 0.287 (±0.02) a | 5.317 (±0.31) ab |
| BR + OM | 20.62 (±2.21) ab | 17.11 (±1.77) b | 8.020 (±0.65) a | 1.420 (±0.07) a | 0.473 (±0.02) a | 4.333 (±0.71) a | 0.770 (±0.05) a | 0.270 (±0.01) a | 5.567 (±0.16) a |
| BR + CQ | 18.11 (±7.5) ab | 16.56 (±3.59) b | 7.573 (±0.58) a | 1.427 (±0.05) a | 0.497 (±0.06) a | 3.767 (±0.52) a | 0.740 (±0.05) a | 0.260 (±0.04) a | 5.150 (±0.28) ab |
| COTROL | 8.91 (±4.03) b | 7.89 (±2.58) c | 7.063 (±0.64) a | 1.373 (±0.11) a | 0.457 (±0.03) a | 3.417 (±0.59) a | 0.720 (±0.03) a | 0.250 (±0.01) a | 4.883 (±0.29) b |

Different letters mean statistically significant differences (value $p \leq 0.05$) in LSD test.; (± Standard deviation). Statistics analysis from plants evaluated 243 DAS.

### 3.7. Grain Nutrient Content

The treatments inoculated with the *Bradyrhizobium* spp. consortia presented higher macronutrient content (g/plant). BR + BA treatment showed the best results with an increase of 157.5%, 150.6%, 173.1%, 168.5%, 182.9%, and 164.8% on N, P, K, Ca, Mg and S respectively, compared to the control (Figure 4). Regarding the micronutrients content (g/plant), BR + BA treatment was also highlighted, by increasing the content of Na, Zn, Cu, Mn, Fe, and B by 228.6%, 163.7%, 100% 198.9%, 195.3%, and 177.8%, respectively, compared to the control (Figure 5). Plants inoculated with the BR + BA and BR + OM treatments significantly stimulated the uptake of macro (N, P, K, Ca, Mg and S) and micronutrients (Na, Zn, Cu, Mg, Fe, and B) on tarwi plants, showing higher values content (g/plant) on grain compared to the control.

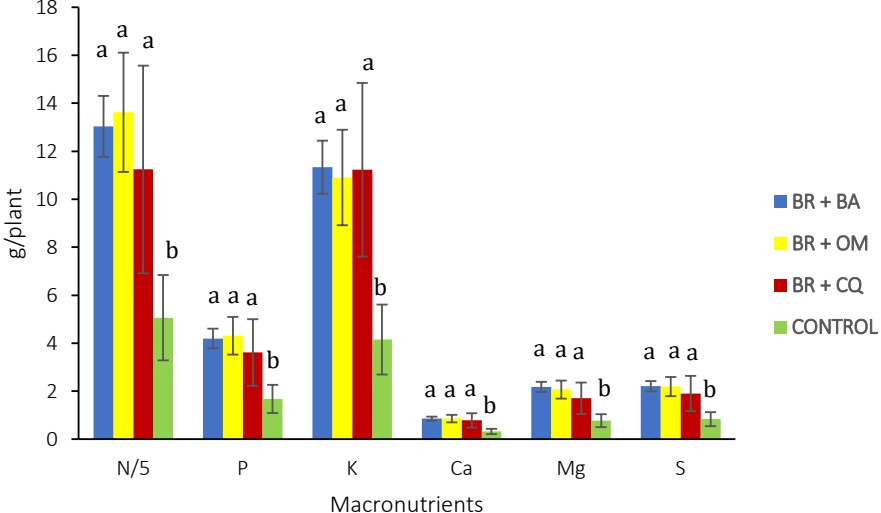

**Figure 4.** Macronutrients content of grains. Different letters mean statistically significant differences (value $p \leq 0.05$) in LSD Test ($\pm$ Standard deviation). (N/5): N value was divided by 5 so that they could be included with the rest of the data in the same graphic.

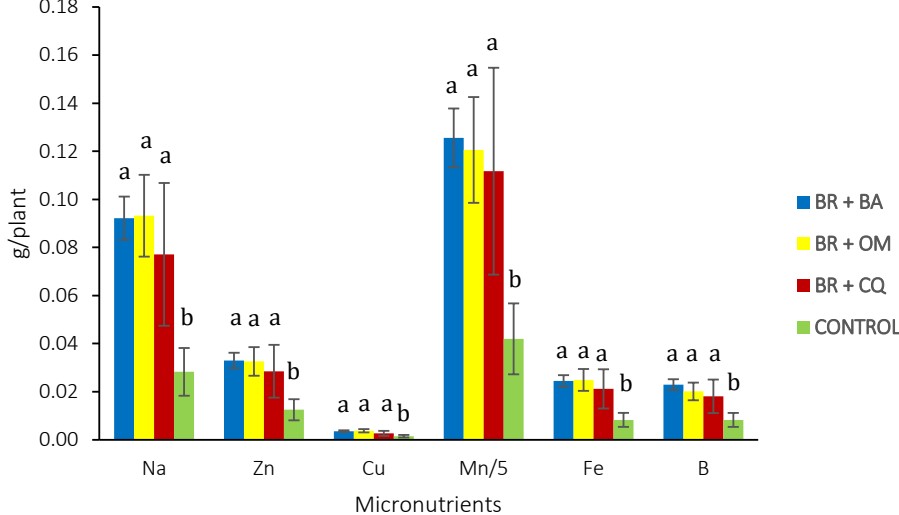

**Figure 5.** Micronutrients content of grains. Different letters mean statistically significant differences (value $p \leq 0.05$) in LSD Test ($\pm$ Standard deviation). (Mn/5): Mn value was divided by 5 so that they could be included with the rest of the data in the same graphic.

### 3.8. Seedling Emergence Percentage from Harvested Seeds

Seedling emergence was evaluated from the sixth to the fifteenth DAS. Fifteen DAS, seeds inoculated with BR + BA treatment showed the best percentage of emergence with

respect to the other treatments with significant differences compared to the control and BR + CQ treatments. Co-inoculation of BR consortia and BA increased the emergence percentage by 62.5% of the seedlings compared to the control (Figure 6). Moreover, the seedling's fresh and dry weight of the BR + BA treatment showed the best values compared to the other treatments (Table 10). BR + BA treatment increased the fresh and dry weight by 97.9% and 16% more, compared to the control.

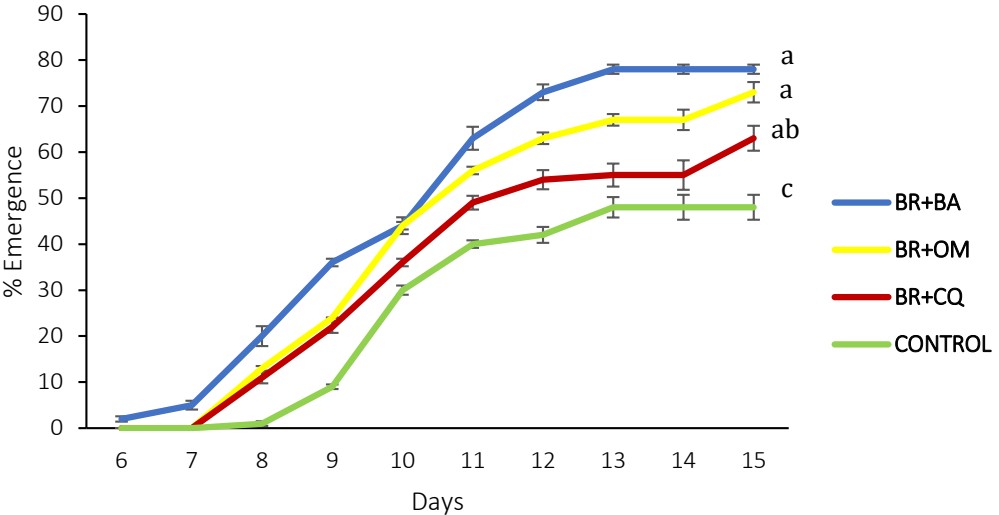

**Figure 6.** Effect of treatments on the % of emerged seedlings. Different letters mean statistically significant differences (value $p \leq 0.05$) in LSD test.

**Table 10.** Effect of treatments on the fresh and dry weight of the emerged seedling.

| Treatment | Plant Fresh Weight (g) | Plant Dry Weight (g) |
|-----------|------------------------|----------------------|
| BR + OM | 37.08 ($\pm$ 3.95) ab | 16.84 ($\pm$ 0.82) ab |
| BR + BA | 42.80 ($\pm$ 6.66) a | 17.60 ($\pm$ 1.85) a |
| BR + CQ | 32.25 ($\pm$ 6.02) b | 16.37 ($\pm$ 1.53) ab |
| CONTROL | 21.63 ($\pm$ 5.23) c | 15.17 ($\pm$ 1.13) b |

Different letters mean statistically significant differences (value $p \leq 0.05$) in LSD Test. ($\pm$ Standard deviation).

## 4. Discussion

The Ancash department is traditionally an area that produces several varieties of tarwi, which are grown annually. Hence, many *Bradyrhizobium* spp. strains were found in the soil. However, the strains used in the current work belong to a selected group obtained in a previous investigation. The strains were selected because they demonstrated promising results in plant growth and seed yield [12,18]. There are only a few publications about PGPR (Plant Growth Promoting Rhizobacteria) and their effects on tarwi crops, despite its nutritional relevance. One of them reported an increase in tarwi's growth and quantity of nodules in the presence of *Bradyrhizobium* spp. inoculant when the experiment was performed under controlled conditions [5]. Likewise, the co-inoculation of *Bradyrhizobium diazoefficiens* USDS110 and *Bacillus subtilis* was reported to increase the number of nodules of soybean (*Glycine max* L.) [19]. However, through the research done during tarwi flowering, plants co-inoculated with *Bradyrhizobium* spp. and *Bacillus* sp. have shown an improvement in aerial growth, but no prominent nodular characteristics. This could be explained because when an experiment is performed under uncontrolled conditions, bacteria colonization is limited, as the plant is exposed to biotic and abiotic stresses. In this case, it would directly affect *Bradyrhizobium* spp. nodulation competitiveness. In this experiment, tarwi plants were affected by a lepidoptera of the Pyralidae family [1] during tarwi's flowering, causing biotic stress (Supplementary Figure S5). This could affect the results, since the sampling was carried out during the flowering. Despite the damage caused by the pest, plants

inoculated with *Bradyrhizobium* spp. showed better nodular characteristics compared to the control.

Regarding the content of chlorophyll, authors state that the amount of chlorophyll in plants is related to its nutritional status and is directly related to its nitrogen content [20]. Plant growth promoter rhizobacteria are reported to improve plant chlorophyll since inoculation increases nitrogen availability in soil [21]. However, it should be noted that the efficiency of this symbiotic relationship depends on biotic and abiotic factors: such as nitrogen deficiency in soils, optimum soil temperature and moisture, and presence of pests, among others. In this experiment, the appearance of the pests during flowering and its effect on nodulation, directly affected the production of chlorophyll, which was shown in the results; where no significant differences were observed between the treatments.

The *Bacillus* sp. BO2 strain has been reported as a biocontroller [7] due to its ability to produce antifungal compounds of hydrolytic enzymes such as proteases, cellulases, and amylases which were used to break down the cell wall components of fungal cells. Siderophores were identified, as well as antimicrobial peptides [22] and other substances that could inhibit the invasion of phytopathogens. On the other hand, field application of *Bacillus* sp. B02 to common bean crops highly reduced the recurrence of the disease caused by *Sclerotinia sclerotiorum* (94%) [23]. This could explain the lower percentage value of severity in the BR + BA treatment since *Bacillus* sp. strain was reported to produce non-volatile antifungal compounds that diminished anthracnose disease damage. Notwithstanding the fact that *Bradyrhizobium* spp. nodulation was affected by the Pyralidae plague; the interaction with *Bacillus* sp. strain helped the plant to have a better response against anthracnose disease.

Regarding the effect on their morpho-agronomic characteristics, it was observed that the BR + BA and BR + OM treatments obtained higher values of leaf height and weight, likewise, they present a greater number of branches (Table 8), greater number and weight of pods per plant. Also, a positive trend was evidenced with respect to the other morpho-agronomic characteristics of the seed and pod (Table 9). These results agree with those reported by [24] who evaluated the inoculation of three strains of *Bradyrhizobium* sp. in *Glyxine max*, finding better characteristics in its morpho-agronomic development. This could be due to the fact that the *Bradyrhizobium* spp. has assimilable nitrogen for the plant [5] and as such, this element promotes greater agronomic development, likewise, the use of *Bacillus* sp. promotes plant protection against damage caused by phytopathogens [7] therefore the plant will be more stable and improves its relationship with its nodules. However, in another investigation [25] that evaluated the co-inoculation of *Bradyrhizobium japonicum* and *Azospirillum brasilense* in soybean cultivars, found greater height in the inoculated plants, but the author affirms that this increase is due to climatic factors, as in the other morpho-agronomic characteristics evaluated, there were no differences. These results may be due to different factors, such as the type of co-inoculant, the variety or ecotype, the agroclimatic conditions, and the presence of diseases and pests, all of which will affect the plant-inoculum relationship. These would explain that only significant differences were found in the number and weight of tarwi pods and seeds in the present investigation.

The research done by [26] affirms that inoculation with rhizobacteria increases plant's absorption of macro and micronutrients, such as Ca, K, Fe, Cu, Mn and Zn. The bacteria make these elements available in the soil, producing organic acids or secreting chelating compounds. Co-inoculation of *Bacillus velezensis* S141 and *Bradyrhizobium* spp., in soybean crop, reported an increase of plant growth and N availability in soil [27]. Likewise, this *Bacillus* sp. strain was found to possess multiple genes that are functionally related to the biosynthesis of auxins and cytokinins [28,29], which could explain, in this case, the advantage of co-inoculation. In the same way the current research showed that co-inoculation with BR + BA increased the content of macro and micronutrient composition in harvested grains. This could be explained by the PGPR abilities of both bacteria such as the production of antifungal volatile compounds [7] and phytohormones [5] which promoted growth and an increase on nutrient content on tarwi seeds. Furthermore, it was observed that nutrient

content of BR + OM was similar to BR + BA treatment. In this experiment, organic matter was made up of remains of previous harvests. These could confer advantages to the plants, because organic matter improves the physical and chemical properties of the soil, promoted *Bradyrhizobium* spp. establishment, and increased plant nutritional balance. The benefit of organic matter application in crops was been reported extensively. [30] claimed that using decomposed cow manure (4 and 8 Tm/ha) significantly increased the number of legumes per plant, and grain weight per plant. In this experiment, treatment with organic matter increased the plant yield compared to the control without fertilizer. [31] also reported that cow manure improves the absorption of N on plants, increasing soy yields. BR + BA treatment also showed the best percentage of emergence. This result could be supported because PGPR are reported to produce phytohormones that improve seeds viability [23]. Seeds with good viability guarantee plant production. So, these inoculants could be incorporated during tarwi sowing to reduce agrochemical demand, which increases production costs and cause soil contamination.

## 5. Conclusions

Results showed that *Bacillus* sp. B02 and the *Bradyrhizobium* spp. consortia formed by LMHZ L8 and LMHZ L3 are effective strains for nodule formation and chlorophyll increase in tarwi var. Andenes crops. Also, they demonstrate that when inoculated together, they have the ability to promote its growth. Improvement of agronomic growth parameters, such as fresh and dry weight of plants and pods are important indicators of crop yield and productivity. In this study, the co-inoculation of these 3 strains demonstrated the ability of the inoculants to induce an increased grain yield, by tripling pod production; almost increasing the nutrient content of grains. This generates not only a greater production, but a product with high nutritional qualities. Additionally, the application of both inoculants diminished the percentage of damage caused by anthracnose, which provides a biocontrol effect to this inoculation technology.

**Supplementary Materials:** The following supporting information can be downloaded at: https://www.mdpi.com/article/10.3390/agronomy12092132/s1, Figure S1: Topographic map obtained from the website Topographic-map.com. Area of the experiment, Tuyu Alto, Marcara, Ancash, Peru; Figure S2: Location of the experiment in Tuyu Alto, Marcara, Ancash, Peru, using images provided by Google Earth, whose location point is -Latitude 9°18′39.17″ S; Longitude. 77°34′29.39″ W, view at 1 km altitude; Figure S3: Sketch of the experiment in Tuyu Alto, Marcara, Ancash, Peru, using images provided by Google Maps, whose location point is Latitude 9°18′39.17″ S; Longitude. 77°34′29.39″ W, view at 50 m altitude; Figure S4: Rainfall, maximum and minimum monthly temperature, presented during the phenological development of tarwi var. Andenes (Fuente: NASA Data Access Power); Figure S5: Presence of turquoise larvae, Lepidoptera of the Pyralidae family, feeding on roots and nodules.

**Author Contributions:** M.M.-G.: Conceptualization; Investigation; Formal analysis; Writing—original draft; Visualization; Data curation. M.M.-Z.: Conceptualization; Methodology; Resources, N.T.: Conceptualization; Methodology; Resources, E.M.: Supervision; Project administration; Resources, K.O.-G.: Writing—review & editing; Validation; Visualization, A.H.-J.: Conceptualization; Methodology, F.C.: Supervision; Project administration; Funding acquisition; Resources; Software, D.Z.-D.: Conceptualization; Supervision; Project administration; Funding acquisition; Writing—review & editing; Validation; Visualization; Data curation. All authors have read and agreed to the published version of the manuscript.

**Funding:** This research was supported by Fondo Nacional de Desarrollo Científico, Tecnológico y de Innovación Tecnológica (FONDECYT) of Peru (N° 079-2018-Fondecyt-BM-IADT-AV and N° 009-2017 projects).

**Institutional Review Board Statement:** Not applicable.

**Informed Consent Statement:** Not applicable.

**Data Availability Statement:** Data will be made available upon request to the corresponding author: dzuniga@lamolina.edu.pe.

**Acknowledgments:** To engineer Robert Quiñones and to biologist Francis Ochoa, who were excellent assistants during data collection.

**Conflicts of Interest:** The funders had no role in the design of the study; in the collection, analyses, or interpretation of data; in the writing of the manuscript, or in the decision to publish the results.

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
