# Peer review of "Co-Inoculation of Bradyrhizobium spp. and Bacillus sp. on Tarwi (Lupinus mutabilis Sweet) in the High Andean Region of Peru"

_agronomy, doi:10.3390/agronomy12092132_

Round 1
Reviewer 1 Report
The study has some interesting and novel aspects which would be of great interest to the international scientific community. However, this manuscript needs significant improvements, both in terms of language and scientific content.
Title
Line 2: “phytopathogenic” – delete.
Abstract
Line 14: “Bradyrhizobium, Bacillus” – italic, followed by “sp.“
Line 15: “Agrochemical” - broad term, be more precise.
Line 18: “plant’s growth” – vegetation.
Line 18-20 – Unclear. Instead of listing the individual parameters in the methodological part of the abstract, it would be better to report the effect of treatments.
Line 21-22 – “Plants co-inoculated with the BR-BA consortium showed the highest values in each parameter evaluated”- not a specific effect, rewrite.
Line 22-23 – “It is concluded that bacterial inoculations improve tarwi growth” – to vague; proper conclusions based on the new findings made in the study are missing.
Line 24: “L. mutabilis” – wrtite full name, italic; “Bacillus”, “Bradyrhizobium” – italic.
Authors should define abbreviations (for example: masl, CI).
Introduction
Line 29-30: where?
Line 38: “Bacillus” – italic.
Line 44: add “sp.“ after “Bradyrhizobium” and “Bacillus”
Line 44: “in” – “on”
Materials and Methods
In Material and Methods, it is noticed insufficient information about Bradyrhizobium sp. and Bacillus sp. regarding its origin, methodology of isolation and identification. On what basis this strain/isolate was selected for the study? Have any characteristics of this strain/isolate already reported in other publications and are selected bacterial strains commercially used? The origin of tarwi variety (Andenes) is not stated.
Line 63, 64: add “sp.“ after “Bradyrhizobium”
Line 65: “LMC” – “YMB”
Line 66: “rotating incubator” – manufacturer, country, state
Line 74: “Bacillus” – italic; add dot after “sp”
Line 75: “phytopathogen” - delete
Line 71-73, 78-79: explain how the inoculation was performed; include more information about agricultural soil used for inoculation.
Line 83-85: include more information about the experiment such as number of rows, row spacing, sowing density
Line 86: include more information about organic matter and agrochemical - what, how much and how exactly they were applied?
Line 91-93: same sentence is repeated (line 93-96); also, methodology for soil properties is missing; define “CIC”.
Line 99: SPAD-502 – country, state of manufacturer?
Line 105: why only three plants per treatment per block?
Line 107-108: manufacturer, country, state
Line 117-118: Colletotrichum gloeosporioides – include more information about pathogen regarding its origin, methodology of isolation, identification and pathogenicity
Line 126, 191, 192, Figure 4: “Kg/ha” – kg/ha
Line 128: “macro and micro-nutrients” – “Macro- and micronutrients”
Line 128-129: “An amount of 100 g of seeds with 2 repetitions of a composite sample of the three repetitions per treatment were used for these analysis” – unclear, rewrite.
Line 135: “agricultural soil” - include more information about agricultural soil used for germination test.
Line 142-143: how many plants were analyzed?
Line 144: delete “:” after “Statistical analysis”
Line 147: Why LSD test was used for yield, and for all other parameters Tukey?
Results
Line 150-151, 156… : “Bradyrhizobium consortium” - What is this treatment? All three treatments contain BR.
Line 153: Table 5 - the difference between control and BR + CQ treatment is too big, are you sure there were no significant differences?
Table 5: explanation for vigor results is missing, define “very good, good, and regular”bellow table 5 (table 3 is unnecessary)
Table 6: check the order of treatments in the table (and values in text); the difference between control and treatments is too big - it is not clear how you did not observed statistically significant differences with these values and for these parameters.
Line 164: write full name of pathogen
Line 166: if you compare the treatments with the control, do not repeat values from the table, for example reduction obtained by BR + CQ in relation to control is 9.12% (60 DAS)
Line 171: table 7 – delete “T1-T4”; “days” – days after sowing
Line 173: “figure 3” – “Figure 3”
Figure 4, Figure 5, Figure 6: explanation for bars is missing
Figure 4: “Kg/ha” in “yield Kg/ha” is unnecessary
Table 8: Height of what? Are these parameters mentioned in the methodology?
Line 206-207: be more detailed
Table 9: Are these parameters mentioned in the methodology? Letters of significance are missing for most parameters.
Figure 5: It seems like some bars are missing. “N/5”, “Mn/5” – explain.
Line 244: “50 das” – 50 or 15?
Figure 6: “Percentage” of? (6B), equalize with 6A; letters of significance are missing in figure 6B.
Discussion
Line 299: reported by?
Line 301, 332: “Bradyrhizobium” – italic
Line 328: “incregggase” – increase?
Line 303: “Bacillus” – italic
Line 336: “re-duce” – reduce; farmers’ – delete
Conclusions
I would like to see a stronger conclusion. Proper conclusions based on the new findings made in the study are missing.
Line 339, 343: “Bacillus”, “Bradyrhizobium” – italic
Line 343: add “sp.” after “Bradyrhizobium”
Line 345: “sus-tainable” - sustainable
Author Response
I have revised my manuscript and submitted a new vesion

Reviewer 2 Report
This work shows the improvement of agronomic parameters of a plant of agronomic interest such as tarwi (Lupinus mutabilis) growing in high altitudes and poor and acid soils, when plants are inoculated with two strains of Bradyrhizobium and one of Bacillus.
The work is interesting but needs to be revised. The tables should be reorganized, the standard error (SE) in the results and is some Figures should be added and the number of repetitions and biological replicates.
The bibliography should be more abundant with a greater number of references and also citing works from higher impact journals.
Suggestions for authors
Title: specify Bacillus (not antagonistic bacteria) and Bradyrhizobium sp., e. indicate that co-inoculation improves growth.
Abstract:
It should be mentioned that tarwi is Lupinus mutabilis.
Line 17 BA was inoculated 60 days after sowing , five times every 30 days
Line 22 BR-BM and BR-CQ show higher values than BR-BA in some parameters!!!!
Line 39, line 283, Bacillus B02 , Is it active against anthracnose caused by Colletrichum?
Line 65, Although a reference is indicated (5), some characteristics of LMHZ L8 and L3 should be mentioned, such as where they were isolated, location, plant, soil type, their efficiency to nodulate, taxonomic position and other PGPR properties.
Line 63, how many ml of the inoculant/plant?, the same in line 78 and Table 1, how many ml of Bacillusinoculum/ plant?
Information of Table 2 can be the legend of Figure 1
Table 2/Fig1, Can the composition of the organic matter be specified?
Can the dosage and frequency of application of agrochemicals be specified?
Information of Table 3 could be included in Table 5.
Information of Table 4 could be included in Table 7.
All tables and Figs (when applicable) should include the values with the standard error (SE) and the number of repetitions and biological replicates
The same typeface must be used in all tables/figs for statistically differences “a” or “A”
In Table 5, are values BR+CQ (236)b not statistically different from CONTROL (58) b???
Table 6 replace “quantity” by number of nodules and specify the number of nodules +SE
Line 166, BR+CQ reduced severity but only significantly compared to T2 and T4. At 110 DAS there is an improvement over anthracnose in the BR treatments (T1, T2 and T3). Does the Bradyrhizobiumconsortium have any antifungal properties?
The data in Figure 4 can be included in Table 8.
Is there really no difference between control and BR+CQ treatment in Figure 4?
Table 8, change "Air weight" (foliar weight??). Please indicate DAS in the Table legend
Table 9, Please indicate DAS in the Table legend and in line 202.
Figs 5 and 6 data should be grouped by macro- or micronutrient, not by treatment.
Line 242 and 244, please write DAS
Line 244, 15 DAS??
The data in Fig. 6B are in Fig. 6A and should not be repeated.
Table 10: It should be noted that the data in Table 10 correspond to the weight of the seedlings 15 days after germination and that the growth conditions are in Material and Methods.
Line 262, There are numerous works where a beneficial effect of bacteria of the genus Bradyrhizobiumand Bacillusin crops is demonstrated, perhaps some more references can be included.
Line 264, Data in Table 6 are very interesting because they show that inoculation achieves the formation of effective red nodules, clearly more efficient than those observed in the uninoculated plant, and this is very important in improving plant yield and nutrition. This should be discussed.
In the discussion, it may be mentioned why uninoculated plants also have nodules (Table 2), is it known what kind of bacteria form them? Is the site from which LMHZ L8 and L3 has been isolated close to where the field trials have been conducted? Is it possible to discuss why the tarwi cultures have not selected efficient endosymbionts? Is the trial site a traditional tarwi cultivation site or has this plant only recently been planted?
Is it also possible to discuss the advantages of inoculation versus the addition of organic matter, since in both cases the results have improved compared to the control and in some cases to the treatment with agrochemicals?
Line 342, It is not correct to say that inoculation suppresses Anthracnose although it does appear to decrease the severity of symptoms.
Supplementary 1, indicate that the map correspond to the trial area....Put the scale
Supplementary 2, Is the trial location Tuyo alto?
Supplementary 3, is a detail of Supplementary 2?. Put the scale
Minor points
Check that "Bacillus and Bradyrhizobium" are in italics throughout the text
Author Response
I have revised my manuscript and uploaded the new version
